# Heterogeneous intercalated metal-organic framework active materials for fast-charging non-aqueous Li-ion capacitors

Nobuhiro Ogihara [1] ✉, Masaki Hasegawa[2], Hitoshi Kumagai[1], Riho Mikita[3] & Naoyuki Nagasako[1]

Intercalated metal-organic frameworks (iMOFs) based on aromatic dicarboxylate are appealing negative electrode active materials for Li-based electrochemical energy storage devices. They store Li ions at approximately 0.8 V vs. Li/Li$^+$ and, thus, avoid Li metal plating during cell operation. However, their fast-charging capability is limited. Here, to circumvent this issue, we propose iMOFs with multi-aromatic units selected using machine learning and synthesized via solution spray drying. A naphthalene-based multivariate material with nanometric thickness allows the reversible storage of Li-ions in non-aqueous Li metal cell configuration reaching 85% capacity retention at 400 mA g$^{-1}$ (i.e., 30 min for full charge) and 20 °C compared to cycling at 20 mA g$^{-1}$ (i.e., 10 h for full charge). The same material, tested in combination with an activated carbon-based positive electrode, enables a discharge capacity retention of about 91% after 1000 cycles at 0.15 mA cm$^{-2}$ (i.e., 2 h for full charge) and 20 °C. We elucidate the charge storage mechanism and demonstrate that during Li intercalation, the distorted crystal structure promotes electron delocalization by controlling the frame vibration. As a result, a phase transition suppresses phase separation, thus, benefitting the electrode's fast charging behavior.

With the rapid growth of the use of lithium-ion batteries in applications such as electric vehicles and smart grids, redox-active organic electrode materials could be considered as candidates to avoid resource risks associated with standard electrode materials such as oxides and carbonaceous materials[1,2]. In addition to continuous research on the design of molecular structures[3,4], studies on organic electrode materials have recently offered functionalization by the self-assembly of molecules as typified by metal–organic frameworks (MOFs)[5,6]. This is because it appears that self-assembly is able to tackle the problems of conventional organic electrode materials, such as physical properties related to electronic conduction[7], chemical stability related to dissolution in electrolyte solution during charging and discharging[8], and low density due to polymerization[6]. In addition, as shown in Supplementary Fig. 1 and Supplementary Table 1, a series of crystalline organic electrode materials can significantly reduce the heat treatment temperature and time compared to standard electrode materials. Since heat treatment is one of the most energy-consuming processes for synthesis[9–11], crystalline organic materials represent a means to achieve reduction in energy consumption during material synthesis, and their utilization could make a significant contribution to future manufacturing and help to decrease its carbon footprint toward carbon neutrality.

We investigated a series of crystalline aromatic dicarboxylates that operate at potentials at ~0.8 V vs. Li/Li$^+$[12–14]. These materials are suitable as negative electrodes for Li-based batteries with good fast-charging performance, which is difficult to achieve with standard

[1]Nobuhiro Ogihara Research Group, Frontier Research Management Office, Toyota Central R&D Labs., Inc., Nagakute, Aichi 480-1192, Japan. [2]Fuel Cell Research-Domain, Emerging Electrification Technology Div., Toyota Central R&D Labs., Inc., Nagakute, Aichi 480-1192, Japan. [3]Secondary Batteries Research-Domain, Emerging Electrification Technology Div., Toyota Central R&D Labs., Inc., Nagakute, Aichi 480-1192, Japan. ✉ e-mail: ogihara@mosk.tytlabs.co.jp

electrode materials such as carbonaceous materials (from 0.5 to 0 V vs. Li/Li⁺) or high-potential electrodes of Ti[15], Nb[16] or Nb-W-based[17] metal oxides (from 1.5 to 2.0 V vs. Li/Li⁺), which are positioned as fast-charging negative electrodes. At the former operating potentials near 0 V vs. Li/Li⁺, safety issues were associated with internal shorts due to Li plating during fast charging[18,19], while the latter higher operating potentials is associated with a limited cell voltage when combined with positive electrodes[20].

The aromatic dicarboxylates form an organic–inorganic layered structure consisting of aromatic π-stacks and a tetrahedral LiO₄ network of carboxylate groups (Fig. 1a and Supplementary Fig. 2) and show reversible Li intercalation via carboxylate anion redox inside the self-assembled structures[12,21]. Thus, we classify a series of these crystalline aromatic dicarboxylates as Li-intercalated MOFs (iMOFs)[12] and systematically investigate their electronic conductivity[7,13], electrochemical properties[21,22], phase transition mechanisms[14], and rate performance as affected by morphology[23] with respect to their crystalline structures. Previous studies revealed that crystals composed of a single organic linker lack comprehensive performance as an electrode material. For example, terephthalate dilithium was first reported as an electrode material[24], but when it was used in a practical battery electrode, the capacity was approximately half the theoretical capacity. In addition, it has large charge–discharge polarization that suggests high resistance[12]. 2,6-Naphthalene dicarboxylate dilithium exhibits high capacity utilization that is almost equal to the theoretical capacity of the practical battery electrode[12], but it also exhibits relatively high charge–discharge polarization and internal resistance[13]. 4,4′-Biphenyl dicarboxylate dilithium also exhibits appealing

theoretical capacity and low internal resistance[14] but it leads to relatively large charge–discharge polarization at the completion of Li deintercalation[13,25].

In this research work, we focus on a multivariate approach[26,27] to iMOFs and their fast-charging performance. As shown in Supplementary Fig. 2, the crystal parameters of these aromatic dicarboxylates have similar values for the lattice constant of the b−c plane, which corresponds to the direction of broadening of the tetrahedral LiO₄ network, while the a-axis, which corresponds to the direction of the length of the organic linker, tends to change. Therefore, our goals were to develop a heterogeneous framework with intermixing and sharing in the b−c plane along with π-stacking homogeneous organic layers and create multivariate iMOFs suitable for fast charging, which cannot be achieved by iMOFs composed of a single organic linker.

To realize this concept, as shown in Fig. 1a, we first searched for the optimal composition by applying machine learning estimation[28] and then synthesized the desired multivariate iMOFs by a spray drying method[23] that allowed solutions to instantly crystallize (Supplementary Fig. 3). Based on the compositions explored by machine learning and shown in Supplementary Fig. 4, the samples synthesized by spray drying had composition molar ratios of naphthalene (Naph, N), biphenyl (Bph, B), and terephthalic acid (Ph, P) of 76:22:2 (SD-NBP(762202)) and 75:7:18 (SD-NBP(750718)). For comparison, we used a single-phase naphthalene framework (SD-Naph) and Naph and Ph in a molar ratio of 75:25 (SD-NBP(750025)). Kinetics based on the phase transition reaction mechanism were discussed based on comprehensive results of material characterization, fast-charging electrode performance, thermodynamic stability, crystal structure

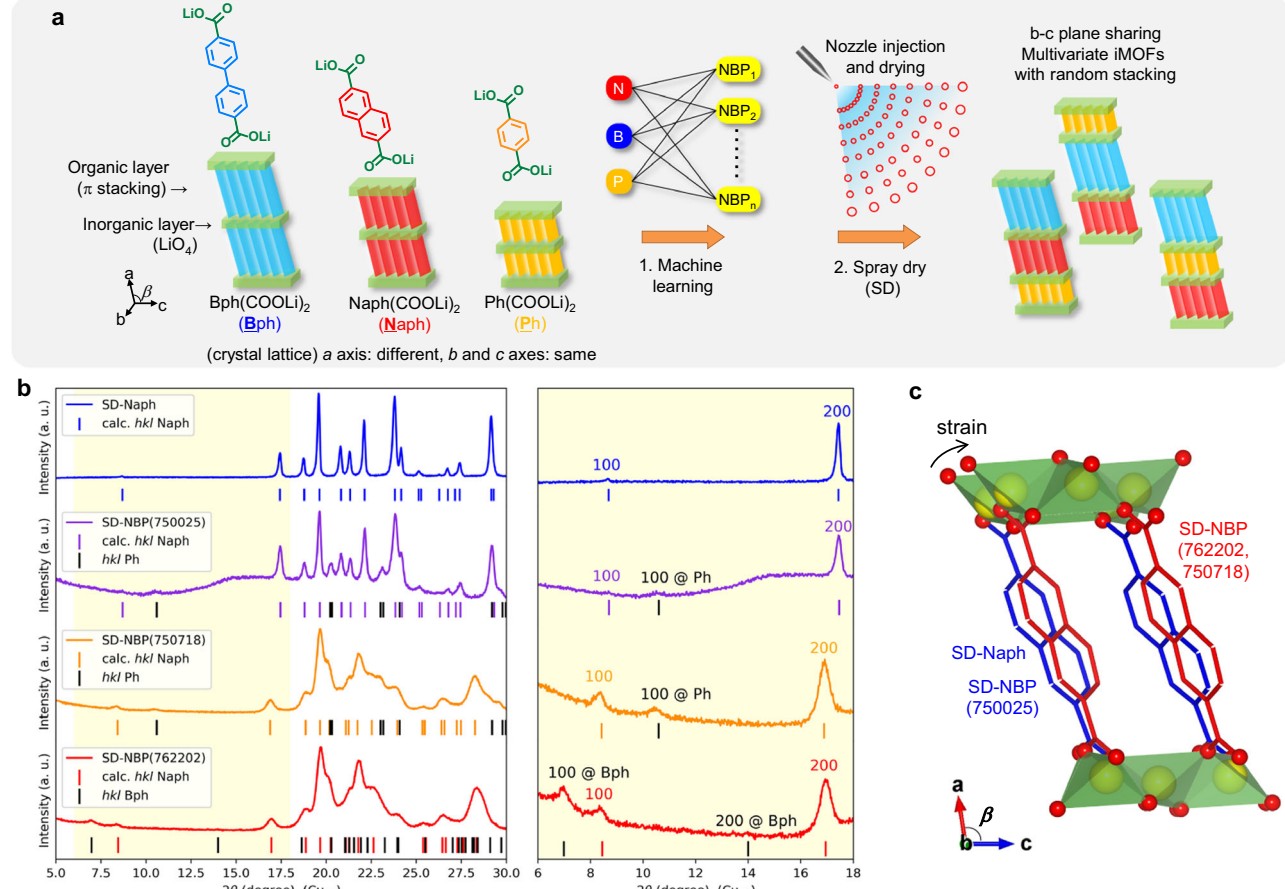

**Fig. 1 | Materials selection, synthesis, and structural characterization.** **a** A concept for the creation of multivariate iMOF materials. **b** Powder XRD patterns. **c** Schematic image of differences in the crystal structure. The red and blue frames are carbon frameworks. Red and yellow spheres represent O and Li, respectively. Green represents the tetrahedral LiO₄ layer.

analysis, and computational analysis to identify the performance enhancement factors of the proposed material. In addition, asymmetric Li-ion capacitors with the best-performing SD-NPB(762202) active material were assembled and evaluated for durability, including cycles and high-temperature storage.

## Results

### Physicochemical characterizations

First, the obtained samples were characterized. According to the results of thermogravimetry and differential thermal analysis (TG-DTA) (Supplementary Fig. 5), the pristine SD-NBP(762202) and SD-NBP(750718) samples contained crystalline water. The samples in SD-NBP(762202) and SD-NBP(750718) show changes in the X-ray diffraction (XRD) patterns before and after vacuum drying at 120 °C that are not observed in the single-phase SD-Naph (Supplementary Fig. 6), and the XRD patterns after drying displayed peaks originating from the space group $P2_1/c$. In addition, these dried samples had a larger specific surface area than the single-phase SD-Naph (Supplementary Fig. 7 and Supplementary Table 2) and an average pore size of ~24 nm (Supplementary Fig. 8 and Supplementary Table 2). Compared to conventional synthesis in solvent and evaporation, spray drying synthesis increases the specific surface area of the biphenyl framework with a single composition[23], whereas that of the naphthalene framework remains unchanged with a single composition but increases only in heterogeneous compositions. Scanning electron microscopy (SEM) measurements confirmed that all samples formed spherical aggregates (Supplementary Fig. 9), which is a characteristic of spray drying synthesis[29]. In addition, after electrode fabrication, SD-Naph was present as cubic particles of 1–2 microns, whereas SD-NBP(762202) and SD-NBP(750718) were nanometric-thin flakes (Supplementary Fig. 10). The latter shape was in good agreement with the previously reported framework of biphenyl from spray drying synthesis[23], and it was in a favorable dispersed state during electrode fabrication.

We carried out comparative XRD analysis to investigate the structure of the multivariate MOFs[26,30–32]. According to Fig. 1b, the XRD patterns after drying for each sample show that SD-Naph exhibits a bulk crystal pattern of the space group $P2_1/c$[8], whereas the heterogeneous samples exhibit a mixture of two phases, which originate from the two highest molar ratio organic linkers contained in the precursor. The respective 100-plane peaks showing *a*-axis regularity depending on the organic linker size were clearly observed in SD-NBP(762202) and SD-NBP(750718), indicating the formation of organic layers by π-stacking interactions of each homoaromatic group rather than that of heteroaromatic mixtures. The 100-plane peak for the naphthalene framework is usually very small or absent due to the extinction law in the space group $P2_1/c$[8], while the 200-plane peak is observed clearly, and the same trend is observed in SD-Naph and SD-NBP(750025). In contrast, the 100-plane peaks for the naphthalene framework in SD-NBP(762202) and SD-NBP(750718) are clearly represented, suggesting a regularity change in the naphthalene framework that inhibits the extinction laws in the *a*-axis direction.

The 200-plane peaks for the naphthalene framework in SD-NBP(762202) and SD-NBP(750718) show lower angles than those in SD-Naph or SD-NBP(750025). The structural parameter results obtained from the XRD patterns (Supplementary Table 3) suggest an increase in the a- and c-axes and a decrease in the b-axis in SD-NBP(762202) and SD-NBP(750718), which is caused by the decrease in the β-angle (Fig. 1c). Even in the heterogeneous samples, the naphthalene framework in SD-NBP(750025) forms a crystal structure similar to that of SD-Naph, whereas that in SD-NBP(762202) and SD-NBP(750718) forms a strained crystal structure different from that of the single-phase SD-Naph. This implies that multiple frameworks interfered with each other in SD-NBP(762202) and SD-NBP(750718), whereas there was no interference in SD-NBP(750025). Therefore, the results for the effect of the extinction law and the strain of the crystal structure suggest the formation of a-axis oriented π-stacked heteroaromatic organic multilayers based on the naphthalene framework was the main component in SD-NBP(762202) and SD-NBP(750718), as shown in Fig. 1a, rather than the formation of crystals of each of the aromatic components without interference, in addition to nanometric-thin flakes formation with high specific surface area associated with the removal of the crystalline water.

### Electrochemical characterizations of iMOFs

The electrochemical behavior of each sample was evaluated using a laminated cell with a Li metal counter electrode. All electrochemical measurements are performed at 20 °C and, the electrolyte is 1.1 mol L$^{-1}$ lithium bis(fluorosulfonyl)imide (LiFSI) salt dissolved in a carbonate-based mixture (LiFSI-based electrolyte), unless specified. The results confirmed a reversible capacity between 200 and 220 mAh g$^{-1}$ per active material at 20 mA g$^{-1}$ (Fig. 2a) with initial coulombic efficiencies of 0.64–0.75 (Supplementary Fig. 11). The reversible capacity corresponded to 2 electron and 2 Li$^+$ ion transfer reactions per organic frame unit, which was equivalent to the theoretical capacity of aromatic dicarboxylates[12]. SD-Naph and SD-NBP(750025) displayed a charge–discharge voltage plateau, while SD-NBP(762202) and SD-NBP(750718) displayed a sloped profile. From the polarization resistance and average potential ($E_{Agv.}$) calculated from the differential capacity analysis $dQ/dV$ plot of each sample (Fig. 2b), compared to SD-Naph and SD-NBP(750025), SD-NBP(762202) and SD-NBP(750718) exhibited approximately 20% less polarization and 60 mV lower average potential, from 0.839 V to 0.777 V (Fig. 2c).

Based on the rate performance of the same cells (Fig. 2d), the charge–discharge curves for SD-Naph and SD-NBP(750025) showed a large profile change with higher current rates, which corresponded to a decrease in rate performance, whereas those of SD-NBP(762202) and SD-NBP(750718) were smaller. In particular, SD-NBP(762202) exhibited favorable capacity retention of more than 85% from 200 mAh g$^{-1}$ at 25 mA g$^{-1}$ to 170 mAh g$^{-1}$ at higher rate of 400 mA g$^{-1}$, the latter specific current corresponding to 30 min of charging (Fig. 2e). The trend in the SD-NBP(762202) results implies the rate performance outlook is expected to be more than 80% of its capacity for 15 min of charging (Fig. 2f), which meets the U.S. Department of Energy's target for extreme fast charging (XFC)[33–35]. Furthermore, as shown in Supplementary Fig. 12 and Supplementary Table 4, the proposed material shows favorable fast charging performance at high loading weights compared to reported MOF electrodes, meaning improved performance at practical electrode loading weights (2.5–3.0 mg cm$^{-2}$)[21]. In terms of cycling characteristics, SD-NBP(762202) and SD-NBP(750718) demonstrate improved capacity retention compared to SD-Naph and SD-NBP(750025) (Fig. 2g and Supplementary Fig. 13), and their crystal structures were maintained before and after cycling (Supplementary Fig. 14). These results suggest that SD-NBP(762202) and SD-NBP(750718) could provide fast charging performance and cycle stability. In previous studies, a single biphenyl framework prepared by spray-dry synthesis found to exhibit high rate properties and significant changing polarization at the completion of Li deintercalation[23], whereas in this study, SD-NBP(762202) exhibited characteristics of charge–discharge behavior without changing polarization.

### Electrochemical testing in asymmetric Li-ion capacitor configuration

To verify the effect of the proposed negative electrode active materials as devices, the performances of Li-ion based asymmetric capacitors[13,21] combined with activated carbon positive electrodes were evaluated using the same LiFSI-based electrolyte (Fig. 3). Here, SD-NPB(762202) was compared to a single layer biphenyl framework electrode fabricated by spray drying (SD-Bph), which exhibits the lowest resistance in the single-framework iMOFs[23]. In this device, the negative electrode properties have a significant impact on the overall device

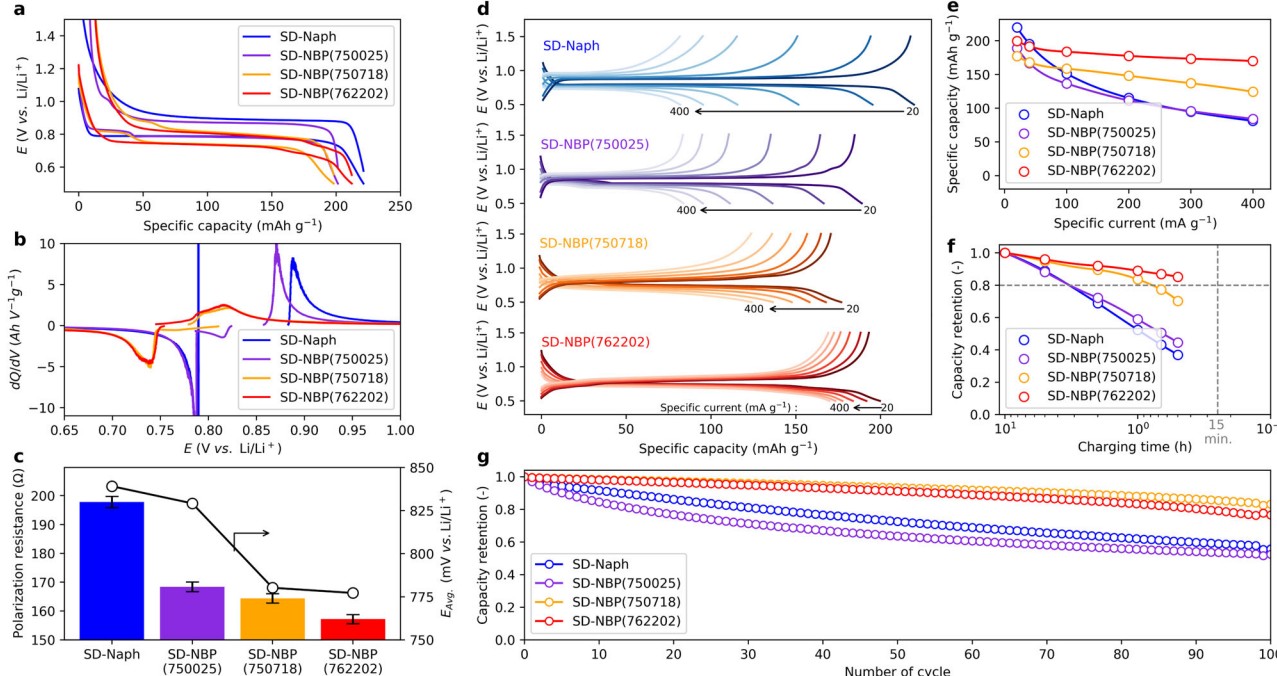

**Fig. 2 | Electrochemical testing in non-aqueous Li metal cell configuration.** **a**, **b** Steady-state charge and discharge potential profiles in Li||iMOF cells at 20 °C (**a**) and their differential capacity $dQ/dV$ plots (**b**). **c** Polarization resistances and average potential ($E_{Avg.}$) for each sample. The polarization resistances were calculated as the difference between the average potential of charge and discharge divided by the applied current. Error bars represent standard deviation. **d** Charge and discharge potential profiles in Li||iMOF cells at different specific currents from 20 mA g⁻¹ to 400 mA g⁻¹ at 20 °C. **e**, **f** Capacity (**e**) and capacity retention (**f**) plots during Li intercalation corresponding to the charging capacity in Li||iMOF cells at each specific current. **g** Capacity retention of the respective Li||iMOF cells over 100 cycles.

performances because the electric double layer interactions with ion adsorption and desorption at the activated carbon positive electrode are sufficiently fast that negative electrode kinetics is the rate-limiting factor[36]. In addition, the overall thermal stability of the device also depends on the negative electrode performance. The negative electrode is pre-doped with Li to utilize the adsorption capacity of both FSI-anions and Li⁺ ions at the activated carbon positive electrode and the low-resistance capacity region at the negative electrode[37,38]. As mentioned in the introduction, SD-Bph exhibits large charge–discharge polarization at the end of Li deintercalation[13,14,25], resulting in a design with a limited capacity range, while SD-NPB(762202) can be designed with a wide capacity range (Fig. 3a, b). The asymmetric hybrid capacitors fabricated with each electrode displayed the desired charge storage response utilizing adsorption of both FSI-anions and Li⁺ ions (Supplementary Fig. 15), and also confirmed both high capacity (Fig. 3c) and low resistance (Fig. 3d) in the SD-NPB(762202)-based cell. The SD-NPB(762202)-based cell displayed better capacity retention of 91% over 1000 cycles at 1 mA cm⁻² compared to that of SD-Bph (Fig. 3e and Supplementary Fig. 16). Furthermore, in the long-term storage characteristics at 60 °C, the SD-NPB(762202)-based cell exhibited favorable capacity retention of more than 70% at a long duration of 300 h at 0.15 mA cm⁻², compared to 50% for the SD-Bph based cell (Fig. 3f and Supplementary Fig. 17). These results mean that the proposed multivariate MOFs not only exhibit low resistance and wide utilization performance but also improve thermodynamic stability. These performance improvements of the proposed materials can be attributed to extrinsic and intrinsic factors[39], related to reduced Li diffusion paths due to nanosize morphology and strained crystal structure, respectively.

## Kinetic investigations on the charge storage mechanism

The scan rate dependence during cyclic voltammetry (CV) was evaluated using the Li metal cells to understand the observed rate

performance from the detailed kinetic behavior related to extrinsic factor due to morphological changes caused by spray drying synthesis. In the peak current for anodic oxidation ($I_{p,a}$) and cathodic reduction ($I_{p,c}$) in CV responses (Fig. 4a), which correspond to Li deintercalation and intercalation reactions, respectively, at different scan rates ($v$), according to Eq. (3) (see "Methods" for details), all plots of the product of peak current and the square root of the scan rate ($I_p v^{1/2}$) show a linear relationship versus $v^{1/2}$ (Fig. 4b), allowing us to determine the fast surface reaction process or the diffusion-limited reaction process from $k_1$ and $k_2$ as coefficients of surface reaction and solid diffusion-limit contributions, respectively[40,41]. The results for $k_1$ and $k_2$ show an increase in $k_1$ and a general decrease in $k_2$ in the order of SD-Naph, SD-NBP(750025), SD-NBP(750718), and SD-NBP(762202) (Supplementary Fig. 18). The calculated value for $k_1/k_2$ tends to be large, according to the above order (Fig. 4c). The anodic $k_1/k_2$ suggests surface reaction dominance and is correlated with capacity retention at 400 mA g⁻¹ for high rate performance in Fig. 2f (Fig. 4d), indicating that kinetic change leads to the observed fast charging properties. This suggests that the kinetic behavior in the above order of the series of materials shifts from the diffusion-limit reaction to the nondiffusion-limit reaction, indicating a fast surface reaction possibly associated with the nanosize of the active material.

Lithium diffusion inside the electrode active material was evaluated using the galvanostatic intermittent titration technique (GITT)[42] to investigate the kinetic factors in detail. The linearity of the potential change with respect to the square root of time ($t^{1/2}$) during current application in GITT is a necessary condition for calculating the diffusion coefficient for Li⁺ ($D_{Li}$, See "Methods" for details). With respect to the coefficient of determination ($R^2$) for that linearity, as shown in Fig. 4e, the calculated values of $R^2$ in GITT for all the samples showed that the R² values for SD-Naph and SD-NBP(750025) were low, whereas those for SD-NBP(762202) and SD-NBP(750718) were close to 1. The details of the normalized transient potential change (ΔE) during

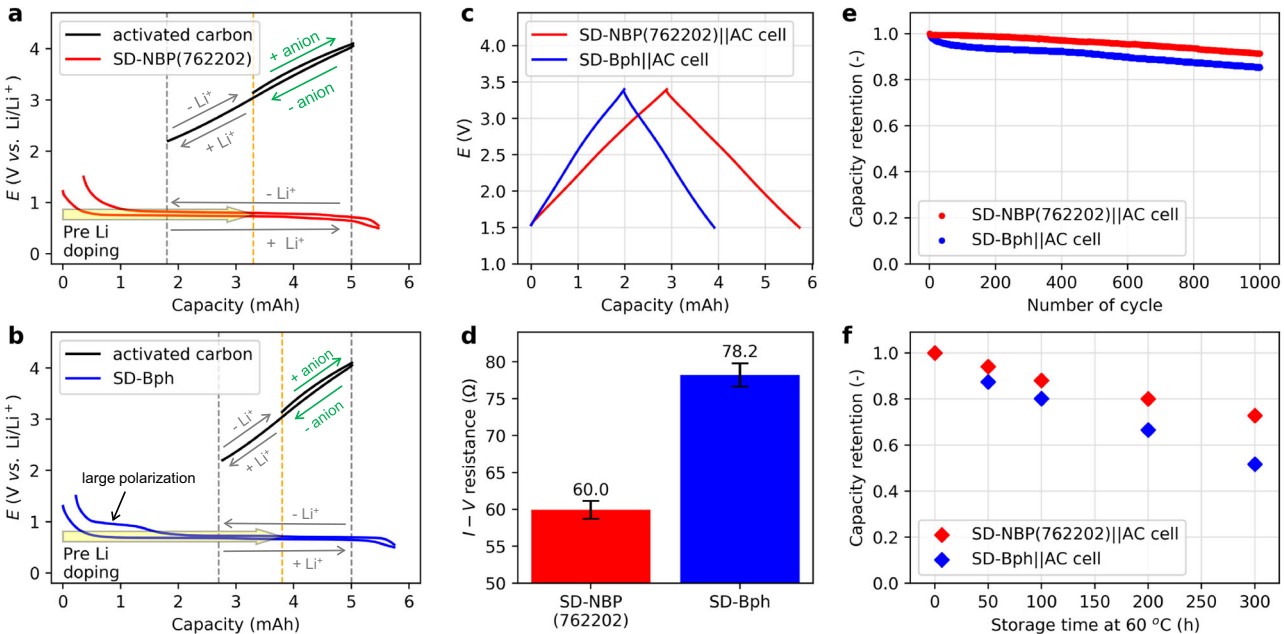

**Fig. 3 | Electrochemical testing in non-aqueous Li-ion capacitor configuration. a**, **b** Design of an asymmetric capacitor with activated carbon (AC) positive electrode and range of capacity utilization due to internal resistance in SD-NPB(762202) (**a**) and SD-Bph (**b**) negative electrodes with pre-lithiated treatment at 20 °C. **c**, **d** Initial charge–discharge curves (**c**), I–V resistance at 0.15 mA cm⁻² (**d**) at 20 °C.

The I–V resistances were calculated as the difference between the average voltage of charge and discharge divided by the applied current. Error bars represent standard deviation. **e**, **f** Capacity retention after 1000 cycles at 20 °C and 1 mA cm⁻² (**e**), and capacity retention at 0.15 mA cm⁻² at 20 °C after storage at 60 °C (**f**) for each cell.

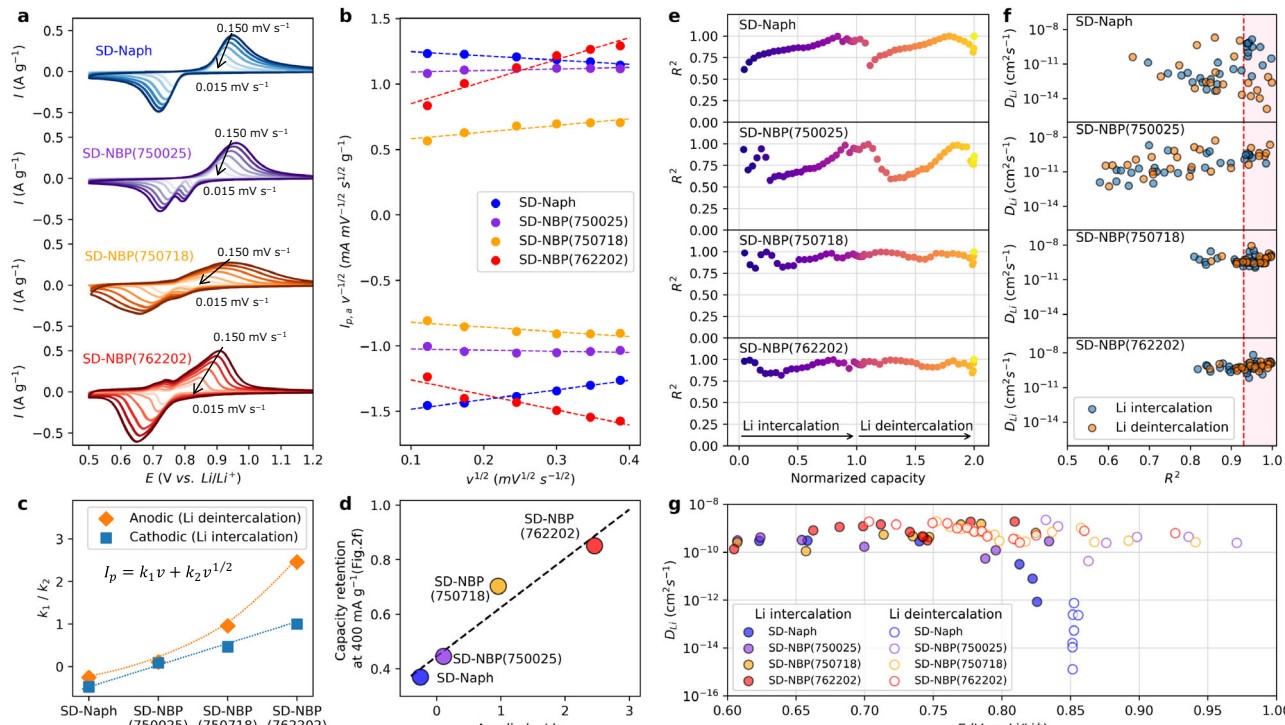

**Fig. 4 | Electrochemical analysis. a** Cyclic voltammograms for Li||iMOF cells. **b** Plots of $I_p$ $v^{-1/2}$ vs. $v^{1/2}$ from 0.015 mV s⁻¹ to 0.150 mV s⁻¹ for Li||iMOF cells. **c** $k_1/k_2$ values for anodic oxidation and cathodic reduction in the CV curves obtained from (**b**) for each sample. **d** Relationship of capacity retention at the specific current in 400 mA g⁻¹ in Fig. 2(f) to the anodic $k_1/k_2$ for each sample. **e** Plots of determination in the linear approximation ($R^2$) of the potential change vs. the

root of time ($t^{1/2}$) during current application in GITT measurements during Li intercalation (0.0–1.0) and deintercalation (1.0–2.0) for each sample vs. normalized capacity. **f** Plots of $D_{Li}$ obtained by GITT vs. $R^2$ for each sample. The areas in pink are treated as solid-solution reaction mechanism areas for $D_{Li}$. **g** Comparison of the plot of $D_{Li}$ vs. the potential for each sample. $D_{Li}$ are the values in the pink area in (**f**).

current application in GITT (Supplementary Fig. 19) show that SD-Naph and SD-NBP(750025) exhibited a large deflection profile that resulted in a flat potential, whereas SD-NBP(750718) and SD-NBP(762202) exhibited a slope-like potential profile. These differences in potential behavior led to different $R^2$ values. According to the relationship between the potential at the Li counter electrode ($\Delta\phi_{Li}$) and the chemical potential of the Li in the intercalation compound ($\Delta\mu_{Li}$) in the Nernst equation (see in Eq. (6) and "Methods" for details), the potential change contributed by $\Delta\mu_{Li}$ reflects the phase transformation of the reaction system[43] and also affects the GITT in short-time pulse measurements, and $R^2$ distinguishes the sloping or flat behavior of the potential change, suggesting a decision indicator for a solid solution or two-phase-coexistence reaction, respectively. This guideline can also be explained by $\Delta E$ vs. $t^{1/2}$ and $R^2$ of GITT for $LiNi_{1/3}Co_{1/3}O_{1/3}O_2$ and $LiFePO_4$ electrodes, which represent typical solid-solution and two-phase-coexistence reactions, respectively (Supplementary Fig. 20). Thus, the $R^2$ values in SD-NBP(762202) and SD-NBP(750718) imply the solid-solution reaction as the phase transition mechanism.

The regions of $D_{Li}$ that can be regarded as solid-solution reactions were investigated from the relationship between $R^2$ and $D_{Li}$ calculated using Eq. (5) in "Methods" for all steps in the GITT (Fig. 4f). The data plots for SD-Naph and SD-NBP(750025) were widely dispersed, while those for SD-NBP(762202) and SD-NBP(750718) were concentrated. The values of $D_{Li}$ were plotted against the potential for the region where the solid-solution reaction mechanism was identified by $R^2$ higher than 0.93, as suggested in the literature[14] (Fig. 4g). As a result, there is a minimum in the $D_{Li}$ values at the potential corresponding to the peak of the $dQ/dV$ curve in Fig. 2b, suggesting that lithium ordering energetically anchored $Li^+$ to the sublattice site and increased the activation barrier for $Li^+$ diffusion[14,44,45]. At these minimum $D_{Li}$ values, those of SD-NBP(762202) and SD-NBP(750718) ($10^{-10}$–$10^{-9}$ cm$^2$ s$^{-1}$) were two to three orders of magnitude higher than those of SD-Naph and SD-NBP(750025) ($10^{-14}$–$10^{-12}$ cm$^2$ s$^{-1}$), similar to previous results obtained by nanosized active materials[23]. Also, these values are comparable to those of conventional negative electrode materials, including graphite ($10^{-11.9}$–$10^{-9.9}$ cm$^2$ s$^{-1}$)[46], $Li_4Ti_5O_{12}$ ($10^{-16}$ cm$^2$ s$^{-1}$)[47], and Nb-W-based oxide materials ($10^{-14}$–$10^{-9}$ cm$^2$ s$^{-1}$)[17,48].

## Phase transition mechanism

To investigate the phase transition mechanism, detailed crystallographic changes as a function of capacity normalized by reversible capacity ($\alpha$) in SD-Naph and SD-NBP(762202) during the Li intercalation process were measured from ex situ synchrotron XRD using samples electrochemically prepared at each Li ratio (Fig. 5a, b). In the XRD pattern of SD-Naph, two phases with 100-plane peaks for the naphthalene framework, corresponding to pristine and Li intercalation states, coexist after $\alpha = 0.1$ (Fig. 5c). This trend of two-phase coexistence is confirmed in the higher angle XRD patterns (Supplementary Fig. 21) and in good agreement with previously reported changes in the bulk naphthalene dicarboxylates[12]. In contrast, in the XRD pattern of SD-NBP(762202), the 100-plane peak in the pristine state is no longer visible after $\alpha = 0.5$, while that in the Li intercalation state appears without the presence of the two peaks (Fig. 5d), indicating suppression of phase separation.

The Li intercalation in SD-Naph exhibits a flat potential profile with a slight potential hysteresis of ~15 mV (Fig. 5a and Supplementary Fig. 22), reflecting the change in $\Delta\mu_{Li}$ caused by the phase separation related to spinodal decomposition[49,50], whereas that in SD-NBP(762202) exhibits a sloping potential profile without hysteresis (Fig. 5b). These behaviors are in good agreement with the predicted changes in the potential profile during the transition from the phase-separated to the solid-solution regime[51]. The observed kinetic change in SD-NBP(762202) is attributed to the effect of the phase transition, which was faster than nucleation[49], with significant diffusivity enhancement.

Next, we discuss the phase transition mechanism leading to fast charging in terms of electron hopping conduction. Based on the results of estimates of unit cell parameters obtained from the results of the ex situ synchrotron X-ray diffraction patterns (Supplementary Fig. 23 and Supplementary Table 5), Li-intercalated SD-NBP(762202) exhibits crystallographic distortions in the main Naph-based framework after Li intercalation, resulting in a reduced lattice size along the $b$-axis, which corresponds to the π-stacking direction of naphthalene. This π-stacking direction corresponds to the direction of main intermolecular electron hopping conduction during Li intercalation[7]. The electronic hopping rate between neighboring molecules ($W$) can be shown as follows[52,53]:

$$W = \frac{V^2}{h}\left(\frac{\pi}{\lambda k_B T}\right)^{1/2}\exp\left(-\frac{\lambda}{4k_B T}\right) \tag{1}$$

where $V$, $\lambda$, $h$, $k_B$, and $T$ are the transfer integral associated with a particular electron level, the reorganization energy (which is defined as the energy change associated with the geometry relaxation during charge transfer), Plank's constant, Boltzmann constant, and temperature, respectively. $V$ is related to the energy partitioning at the electron level when going from an isolated molecule to an interacting molecule and is given by the splitting of the highest occupied molecular orbital (HOMO) and lowest unoccupied molecular orbital (LUMO) levels associated with that interaction[54,55]. Higher HOMO (LUMO) bandwidths increase the transfer integral, resulting in high electron hopping conduction. Since a reduction in the distance for electron hopping between neighboring molecules increases the electron splitting amplitude at the HOMO and LUMO levels and promotes hopping conduction[54], the observed reduction of the distance between naphthalene frames caused by structural distortion suggests enhanced electron hopping conduction in solids.

## Vibration behavior in organic frameworks

To examine intrinsic structural factors affecting the observed phase transition mechanism, the vibrational state of the molecules inside the framework was examined from a combined analysis of Raman spectra for each fully lithiated sample and predictions of their vibrational modes by first-principles phonon calculations (Supplementary Figs. 24 and 25). The combined results revealed that the peaks corresponding to bending vibrations perpendicular to the naphthalene plane ((i) as a representative vibration mode in Fig. 6a) were not clearly observed for the fast charging samples SD-NBP(762202) and SD-NBP(750718), whereas the peaks corresponding to stretching vibrations parallel to the naphthalene plane ((iv) as a representative vibration mode in Fig. 6a) remained unchanged (Fig. 6b). The former Raman spectral difference means the suppression of structural fluctuations leading to the persistent naphthalene planarity in SD-NBP(762202) and SD-NBP(750718). The bending vibration perpendicular to the naphthalene plane is a motion that disturbs the aromatic planarity, which negatively affects the electron transfer because the planarity is related to the delocalization of π electrons[56–58]. The absence of the bending vibrations may be attributed to the strained structure observed in SD-NBP(762202) and SD-NBP(750718) and maintains π-electron delocalization by reducing these planarity-disturbing bending vibrations, thereby contributing to the enhancement of electron transfer avoiding phase separation during Li intercalation that leads to the observed fast charge performance with high-temperature stability.

## Discussion

Multivariate material design is an interesting approach for finding crystalline aromatic dicarboxylates for use as electrode active materials. Our results reveal two fusion effects of extrinsic and intrinsic factors: control of nanosize morphology formation in spray-dry synthesis and framework distortion in the optimal composition in

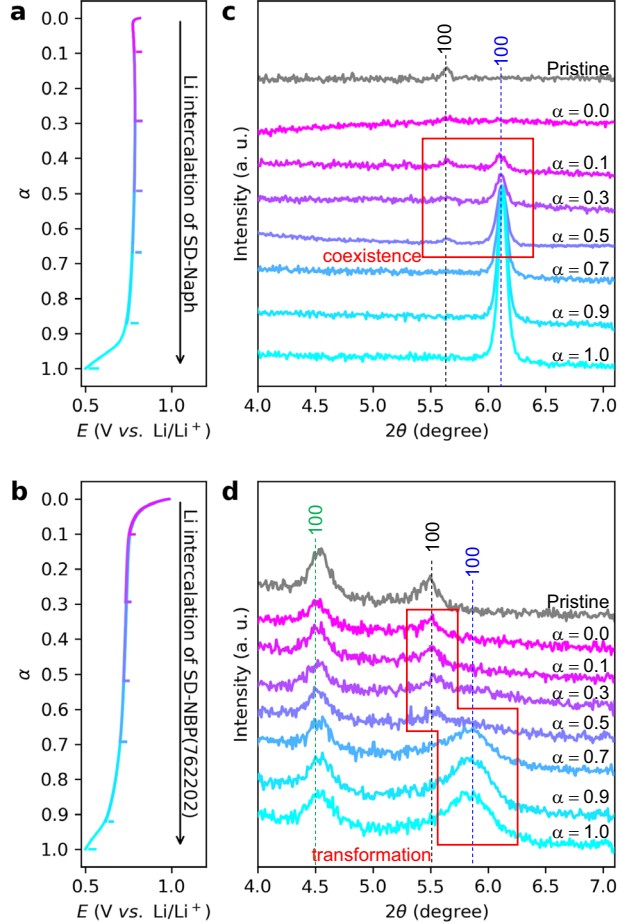

**Fig. 5 | Ex situ XRD patterns during the Li intercalation reaction.**
**a**, **b** Superimposed profiles of potential-normalized Li ratio (α) during Li inter-
calation of the respective SD-Naph (**a**) and SD-NBP(762202) (**b**) for the preparation
of ex situ XRD samples. The α was defined as the capacity ratio normalized by the
reversible capacity. **c**, **d** Variation in XRD patterns of SD-Naph (**c**) and SD-
NBP(762202) (**d**) with respect to the normalized Li ratio. Black and blue numbers
indicate the indices of the pristine and Li intercalation phases in the naphthalene
frame, respectively. The green number indicates the pristine index in the biphenyl
framework.

multivariate MOFs using machine learning, respectively. The former
effect leads to improved surface reaction and Li diffusion based on the
results of the scan rate dependence in CV and GITT results, while the
latter affects the avoidance of phase separation due to enhanced
electron transfer, suggesting molecular vibration control effects as
their mechanism based on the results of potential profiles, crystal
structure changes during Li intercalation, and Raman spectra in lithi-
ated samples. These effects resulted in faster charging performance,
an expanded low-resistance capacity region, charge–discharge cycle
stability, and improved high thermal stability. Thus, this approach
expands the design possibilities for organic crystalline materials and
could contribute to future sustainability through resource risk avoid-
ance and reduced energy consumption during manufacturing.

## Methods

### Material synthesis

The investigation of the optimal composition in the three-component
organic linker was conducted by machine learning using a prediction
model based on the random forest technique[28]. The objective variable
was reversible capacity and polarization of charge and discharge
related to internal resistance. The explanatory variables were similarity
metrics as Pearson's correlation coefficient from the XRD pattern of

arbitrary composition samples and that of each single phase such as
Ph, Bph and Naph. The hyperparameters were optimized to show the
lowest Root Mean Squared Error for the test data set, with accuracy
improved by tenfold cross-validation. The sample was synthesized via
spray drying (MDL-050, GF Corporation), as shown in Supplementary
Fig. 3, using an aqueous solution comprising 0.2 mol $L^{-1}$ of mixed state
aromatic dicarboxylic acid, 2,6-naphthalenedicarboxylic acid, 4,4′-
biphenyl dicarboxylic acid and terephthalic acid (Tokyo Chemical
Industry Co. Ltd) and 0.44 mol $L^{-1}$ of lithium hydroxide monohydrate
(FUJIFILM Wako Pure Chemical Corporation). The spray amount was
0.4 L $h^{-1}$, and the drying temperature was in the range of 150–200 °C.
The resulting solid was dried under vacuum at 120 °C. The detailed
composition of molar ratios for each sample is shown in Supplemen-
tary Fig. 4. The powders were collected by the cyclone (Cy) and bag
filter (BF) sections, and BF samples were used to evaluate electrode
properties. A single-phase biphenyl framework was also synthesized
using spray drying (SD-Bph)[23] as a comparison for device evaluation.

### Electrode and electrolyte preparation

The electrodes were prepared by coating a dispersion of the
respective sample (74.1 wt%), carbon black (18.5 wt%, Tokai Carbon
Co., Ltd.), carboxymethylcellulose (1.8 wt%, Daicel Fine Chem Ltd.)
and modified polyvinyl alcohol (5.6 wt%, T-330, Gohsenol) in water
onto 10-μm-thick Cu foil (99.9% purity, Fukuda Metal Foil & Power
Co., Ltd.). The loading weights of both electrodes were ~3 mg $cm^{-2}$.
The activated carbon (AC) electrodes were also prepared by a dis-
persion of active material (90 wt%), carbon black (4 wt%), styrene-
butadiene rubber (5 wt%), and carboxymethylcellulose (1 wt%) in
water onto 20 μm-thick Al foil (99.9% purity, Japan Capacitor Indus-
trial Co., Ltd.). The loading weights of both electrodes were
3–4 mg $cm^{-2}$. The LiNi$_{1/3}$Co$_{1/3}$Mn$_{1/3}$O$_2$ (NCM111) and LiFePO$_4$ electro-
des used for comparison were prepared by coating a dispersion
composed of active material (92 wt%), carbon black (5 wt%), and
polyvinylidene fluoride (3 wt%, Kureha Corporation) as the binder in
N-methyl-2-pyrrolidone on aluminium foil. The electrode thickness
was 40–50 μm. A 200-μm-thick Li metal foil (99.8% purity, Honjo
Metal Co., Ltd.) was pressed onto a 10-μm-thick Cu foil and used as
the counter electrode. The electrolytes were made of lithium
bis(fluorosulfonyl)imide (LiFSI) or LiPF$_6$ dissolved in a mixture of
ethylene carbonate, dimethyl carbonate, and ethyl methyl carbonate
(volume ratio = 30:40:30) with a concentration of 1.1 mol $L^{-1}$ (the
water content less than 10 ppm, Kishida Chemical Co., Ltd.). A 25-μm-
thick polypropylene microporous membrane was used as the
separator (Celgard 2500, Celgard LLC).

### Characterization

The pore distribution of the samples was measured by N$_2$ adsorp-
tion (BELSORP-max II, Microtrac Bell). The specific surface area
($S_{BET}$) and pore distribution ($V_{BJH}$) were calculated from the N$_2$
adsorption isotherm using the Brunauer–Emmett–Teller (BET)
and Barrett–Joyner–Halenda (BJH) methods, respectively. The
morphologies of each sample powder and electrode were examined
using scanning electron microscopy (SEM, S-5500, Hitachi, Ltd.,
Tokyo, Japan). Thermal analysis of the samples was performed using
a thermogravimeter-differential thermal analyzer (TG-DTA,
THERMO PLUSEVO II, Rigaku). The measurement temperature
ranged from room temperature to 1000 °C with a temperature
increase rate of 5 °C min$^{-1}$. The X-ray diffraction (XRD) data of the
pristine sample particles and sample electrodes before and after
cycle test were obtained using a Rigaku Ultima IV diffractometer
(Rigaku Corporation) with Cu$_{Kα}$ radiation at 50 kV and 300 mA in
the 2θ$_{CuKα}$ range of 5–30° in reflection mode under an air atmo-
sphere. Raman spectra of samples during Li intercalation were
measured using Raman microscopy (RAMANtouch, Nanophoton
Co., Ltd.). Prior to Raman measurements, Li-intercalated electrode

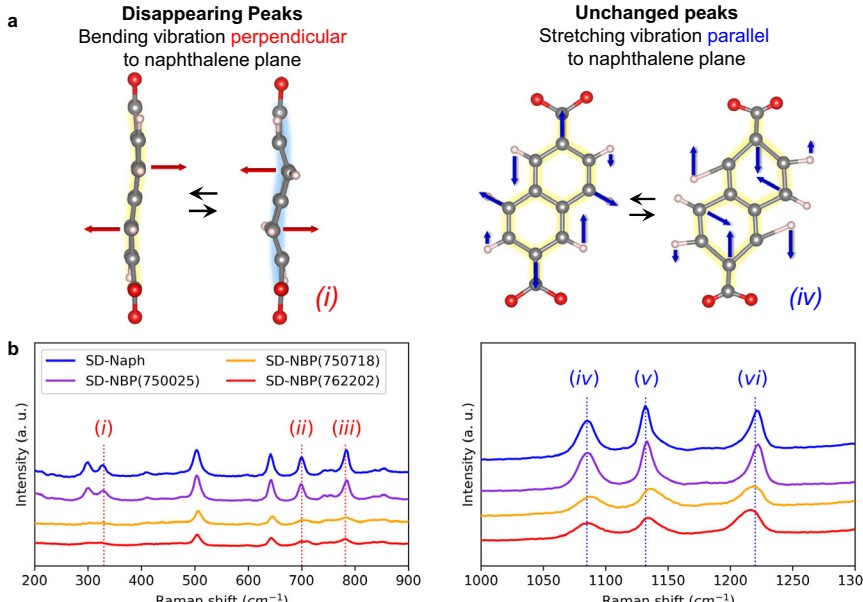

**Fig. 6 | Vibration modes of the framework during Li intercalation.**
**a** Representative vibration modes predicted by phonons using first-principles calculations. Gray, red, and pink spheres represent C, O, and H, respectively. **b** Raman spectra for each fully lithiated sample. Detailed experimental and computational Raman spectra and other vibration modes are shown in Supplementary Figs. 24 and 25, respectively.

was taken from the electrochemical cell in an argon-filled glove box and washed with ethyl methyl carbonate; and the removed electrode was placed in a sealed cell for the measurement. The power of the laser with a wavelength of 532 nm was kept in the 2–10 mW range and measurements were performed for 100–300 s. The ex situ XRD pattern changes for SD-Naph and SD-NBP(762202) electrodes during Li intercalation were obtained using a Debye-Scherrer camera at the BL5S2 beamline, Aichi Synchrotron Radiation Center, Aichi prefecture, Japan (proposal no. 2021D6020). The data were recorded in transmission mode using a PILATUS 100 K detector (DECTRIS AG) with a resolution of 0.6195 Å (calibrated using the standard $CeO_2$ powder), 2θ values of 0–95°, and a step size of 0.01°. The samples for XRD measurements were prepared for the respective Li ratios using a Li‖iMOF cell with the same electrolyte at 20 °C, respectively. Prior to XRD measurements, the capacities of all Li‖iMOF cells for the XRD samples were confirmed to be equal and adjusted to be the respective Li ratios. The prepared electrode was taken from the electrochemical cell in an argon-filled glove box and washed with ethyl methyl carbonate; samples were obtained from the electrode; and the powder was packed into a borosilicate glass capillary tube with an external diameter of 0.3 mm. VESTA software was used to draw the three-dimensional structures of the crystal structures[59].

**Galvanostatic charge–discharge measurement**
The electrochemical properties of Li‖iMOF cells were examined using laminate-type cells with a 10 cm² electrode area assembled by employing a separator filled with LiFSI-based electrolyte (approximately 0.5 mL per cell) in an argon-filled glove box with negligible $H_2O$ and $O_2$ levels below 0.1 ppm (Supplementary Fig. 26). For the galvanostatic charge–discharge measurements to determine the specific capacity and electrochemical reversibility of the sample electrodes, the cells were cycled between 0.5 and 1.5 V (vs. Li/Li⁺) at a specific current of 20 mA g⁻¹ corresponding to the fully charged theoretical capacity of the sample per 10 h for charge/discharge characteristics at low specific current. The polarization resistances were calculated as the difference between the average potential of charge and discharge divided by the applied current. The rate characteristics using the same cell were cycled three times between 0.5 and 1.5 V at various specific

currents from 20 to 400 mA g⁻¹, corresponding to the fully charged theoretical capacity of the sample per 10 h to 0.5 h. All specific currents were calculated for the weight of active material in the electrode. The cycling test using the same cell was cycled 100 times between 0.5 and 1.5 V at a specific current of 100 mA g⁻¹. All electrochemical measurements were performed at an environmental temperature of 20 ± 0.5 °C. To ensure reproducibility, electrochemical characterization was performed at least twice per sample.

**Cyclic voltammetry (CV)**
CV was measured at scan rates of 0.015, 0.030, 0.060, 0.090, 0.120, and 0.150 mV s⁻¹ at the same potential range of 0.5–1.5 V at 20 °C.

**Galvanostatic intermittent titration technique (GITT)**
The GITT was performed using the same cells as in the galvanostatic charge–discharge measurement above. The cells were charged and discharged with a short pulse of a constant current responsible for the transfer of 0.1e⁻ and 0.1Li⁺, followed by interruption for 100 h or until the potential change decreased to <4.5 mV h⁻¹ at 20 °C. This procedure was repeated in the range of 0.5–1.5 V.

**Device evaluation as asymmetric capacitors**
The evaluated asymmetric capacitors were examined using the same laminate-type cells composed of AC-positive and pre-lithiated SD-NBP(762202) and SD-Bph-negative electrodes with separators filled with the same LiFSI-based electrolyte. Prior to galvanostatic charge–discharge measurements of the asymmetric capacitors, as shown in Fig. 3a, considering their respective low-resistance capacitance ranges, SD-NBP(762202) and SD-Bph electrodes were pre-lithiated by discharging laminate-type Li‖iMOF cells up to 65 and 75% of total Li intercalation capacity, respectively. Asymmetric capacitors were then prepared using the respective electrodes taken from the cells and the AC electrodes in an argon-filled glove box. The cells were cycled between 1.5 and 3.4 V at a current density of 0.15 mA cm⁻² corresponding to the fully charged capacity of the cell per 2 h to confirm their initial performance at 20 °C. The cycling test using the same cell was cycled 1000 times between 1.5 and 3.4 V at a current density of 1 mA cm⁻². For high-temperature storage performance, the cells were charged to 3.4 V at a current density of 0.15 mA cm⁻² and

then stored at 60 °C for each specified time. After returning to 20 °C, the charge and discharge capacities were checked between 1.5 and 3.4 V at a current of 0.15 mA cm$^{-2}$.

## Raman spectra calculation

First-principles calculations for Raman spectral prediction were performed by the projector augmented wave method as implemented in the Vienna ab initio simulation package (VASP)[60–63]. We adopted the generalized gradient approximation (GGA) for the exchange-correlation energy and potential and employed the PBEsol functional[64], which is the revised version of the expression suggested by Perdew, Burke, and Ernzerhof (PBE)[65,66]. The cutoff energy for wave functions was set to 600 eV. In the structure optimization, the occupancy of each electronic state was calculated using Gaussian smearing with a smearing width of 0.2 eV. Integration in reciprocal space was performed with a k-point mesh of (7, 13, 9), (9, 13, 7), and (7, 13, 9) for 2,6-Naph(COOLi)$_2$, 2,6-Naph(COOLi$_2$)$_2$ (type 1) and 2,6-Naph(COOLi$_2$)$_2$ (type 2), which were the previously determined crystal structures[12]. Phonon dispersions were calculated by the direct method[67]. A supercell of (1, 2, 1) unit cell was used for all materials. Raman spectra at Raman-active frequencies were calculated by using Phonopy-Spectroscopy[68].

## Kinetic behavior analysis in CV

The redox current response ($I_p$) in CV can be divided into two main contributions: the kinetic behavior of the fast surface reaction process proportional to the scan rate ($v$), which is not solid diffusion-limited, and the solid diffusion-limited reaction process proportional to $v^{1/2}$. When $k_1$ and $k_2$ are defined as coefficients of surface reaction and solid diffusion-limit contributions, respectively, $I_p$ is shown as follows[40,41]:

$$I_p = k_1 v + k_2 v^{1/2} \tag{2}$$

which can be rearranged into:

$$I_p v^{-1/2} = k_1 v^{1/2} + k_2 \tag{3}$$

By plotting $I_p$ for different $v$, it is possible to derive $k_1$ and $k_2$ from the slope and intercept, respectively, based on the relationship in Eq. (3) and estimate the contribution of the fast surface reaction or the diffusion-limited reaction processes.

## Li$^+$ diffusion analysis in GITT

The potential changes during short current pulses and subsequent relaxation during Li intercalation and deintercalation were measured by GITT (Supplementary Fig. 27a). Assuming that the applied current and steady-state open circuit potential change ($\Delta E_s$), as shown in Supplementary Fig. 27b, are sufficiently small, the chemical diffusion coefficient for Li$^+$ ($D_{Li}$) can be determined by the following equation:

$$D_{Li} = \frac{4}{\pi}\left(\frac{m_B V_M}{M_B S}\right)^2 \left(\frac{\Delta E_s}{\tau(dE/dt^{1/2})}\right)^2 \quad (\tau \ll L^2/D_{Li}) \tag{4}$$

where $m_B$, $V_M$, $M_B$, and $S$ represent the mass, molar volume, molecular weight of the material, and surface area of the electrode, respectively. $t$ and $L$ represent the time during which the current pulse is applied and length of electrode material, respectively. When the observed potential changes linearly with respect to the square root of time ($t^{1/2}$) during the application of the current pulse, Li$^+$ intercalation or deintercalation can be treated as a "solid-solution-type" mechanism in which Li$^+$ randomly occupies the intercalated sites. Then, Eq. (4) can be transformed using the

potential change upon applying the pulse current ($\Delta E_\tau$).

$$D_{Li} = \frac{4}{\tau\pi}\left(\frac{m_B V_M}{M_B S}\right)^2 \left(\frac{\Delta E_s}{\Delta E_\tau}\right)^2 \tag{5}$$

In contrast, when the plot of $E$ vs. $t^{1/2}$ shows nonlinearity, Li$^+$ intercalation or deintercalation can lead to the coexistence of two phases ("two-phase-type" mechanism)[45,69]. As shown in Supplementary Fig. 27c, the above classification of the reaction mechanism can be estimated from the correlation coefficient ($R^2$) of the linear approximation of $\Delta E$ vs. $t^{1/2}$ when the current is applied[14,70].

## Potential at the Li counter electrode ($\Delta\phi_{Li}$) in the Nernst equation

$\Delta\phi_{Li}$ can be expressed using the chemical potential of the Li in the intercalation compound ($\Delta\mu_{Li}$) and the reaction electrons ($e$), where $\Delta\mu_{Li}$ is equal to the derivative of the Gibbs free energy of the material ($G$) with respect to the normalized Li ratio ($a$)[43,49].

$$\triangle\phi_{Li} = -\frac{\triangle\mu_{Li}}{e} \left(\triangle\mu_{Li} = \frac{\partial G}{\partial \alpha}\right) \tag{6}$$

In the case of a solid-solution reaction occurring in a single phase, the potential exhibits a smooth sloping profile due to the change in $\Delta\mu_{Li}$, whereas in the case of a two-phase coexistence reaction with a first-order phase transformation, the voltage exhibits a flat profile due to the constant value of $\Delta\mu_{Li}$, which linearly connects the local minima of the Gibbs free energy of the two phases caused by phase separation related to spinodal decomposition[43].

## Data availability

All data generated or analyzed during this study are included in the published article and Supplementary Information and are available from the corresponding authors upon request.

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

## Acknowledgements

The powder XRD measurements were performed at the BL5S2 beamline, Aichi Synchrotron Radiation Center, Aichi prefecture, Japan (proposal no. 2021D6020).

## Author contributions

N.O. conceived, designed, and performed the experiments and prepared the manuscript. M.H. and H.K. performed the spray drying synthesis. N.N. performed the first-principles calculations of Raman spectra. R.M. evaluated the capacitor performance. All authors discussed the results and commented on the manuscript.

## Competing interests

The authors declare no competing interests.
