## [Peer Review File · Nature Communications]

REVIEWER COMMENTS

Reviewer #1 (Remarks to the Author):

This manuscript reports an approach that combines machine learning and solution spray drying techniques to select and produce multivariate crystalline aromatic dicarboxylate intercalated metal-organic frameworks (iMOFs) for serving as Li-ion host negative electrode materials. To convince the concept, naphthalene-based multivariate iMOFs with specific compositions were predicted and prepared. These samples were characterized as nanoflake-like distorted crystals composed of *n*-stacked heteroaromatic multilayers and showed enhanced electrochemical properties, including high-rate capability and cycling stability, as compared to a single aromatic counterpart. Overall, this manuscript is in good shape, and the referee recommends acceptance of this paper in Nature Communications, but not before the referee's concerns listed below are addressed.

1. The message delivered by Fig.1 was clear and straightforward, but Fig.1 also implies that the listed electrode materials are produced at a specific temperature which could be a bit misleading since some of these electrode materials are commercially produced at various temperatures. Taking graphite as an example, there is a huge difference in terms of the production temperature for natural graphite and artificial graphite. The referee would recommend the authors either improve Fig.1 with a temperature range instead of a single value of each material, or state clearly in the caption that these temperature values listed in Fig.1 are specific temperature conditions. The referee would also recommend moving Fig.1 to the Supplementary materials.

One last minor comment related to Fig.1 is that the term "Mxene (Ti₂AlC)" was used to describe MXene materials, however, this could be a bit misleading since Ti₂AlC is the MAX precursor to produce Ti₂CT_x MXene. And the temperature of Ti₂AlC was labeled as 1600 oC. This should be updated since many MAX phases, taking the well-known Ti₂AlC, Ti₃AlC₂, and Ti₃SiC₂ for example, can be produced at a temperature range of 1100-1400 oC (Ref: J Am Ceram Soc. 2021;104:659-690).

2. The authors should state the composition ratios of Naph, Bph, and Ph are depending on weight or molar.

3. More information should be provided on why these three particular ternary Ph-Naph-Bph compositions were selected by the machine learning prediction for further study. Based on what kind of parameters and conditions?

4. The claim that "...a-axis oriented *n*-stacked heteroaromatic organic multilayers based on the naphthalene framework was the main component in SD-NBP(762202) and SD-NBP(750718)..." on Page 8 is one of the key features of these multivariate

iMOFs samples, are there more experimental evidence, such as HR-TEM analysis, to support such claims? And on the same sentence, the authors continued, "...in addition to nanoflake formation due to the high specific surface area associated with the removal of water crystals." What is the role of water in the multivariate crystallines? Some discussion would be appreciated.

5. What is the initial coulombic efficiency of these electrodes?

6. On page 9, the authors wrote: "This result for SD-NBP (762202) implies that it can store more than 80% of its capacity at the 4C rate for 15 minutes of charging (Fig. 3f)..." Did the authors test the SD-NBP (762202) electrode at the 4C rate to support such claims?

7. The authors used the term "during Li intercalation " in captions of Fig.5 and Fig.S15 and Methods to describe the Raman results of these four electrodes. What are the lithiation states of the Raman measurements? If the Raman spectra were recorded based on fully lithiated electrodes, the authors should state it as such.

8. The authors claimed, "The Li intercalation in SD-Naph exhibits a flat potential profile with a slight potential hysteresis of approximately 15 mV (Fig. 6a)..." on page 14. Can the authors point out where this 15 mV has been observed?

9. On page 15, the authors wrote, "SD-NBP(762202) exhibits crystallographic distortions in the main Naph-based framework after Li intercalation, resulting in a reduced lattice size along the b axis,..." The author should make it clear that this "reduced lattice size" is under comparison with Li-intercalated SD-Naph rather than pristine SD-NBP (762202).

10. The Summary on page 16 is too general; adding some specific results would make it more appealing.

Reviewer #2 (Remarks to the Author):

In this manuscript, the authors fabricated the multivariate iMOFs electrodes to realize the reversible and fast Li charging, which are also unique anode materials with high safety. Optimal compositions were obtained based on machine learning. The spray-drying provided a synthetic solution with high efficiency and low energy cost. It is overall an interesting work, however, there are some obvious inconsistencies and major concerns that the authors need to take into consideration.

Firstly, the correlationship of structure-morphology-electrochemical properties is

not strong.

1. The NaPh, BPh and Ph alone have large resistance as electrode materials, but the multivariate iMOFs have small internal resistance and polarization. It seems that the intensity of stretching vibrations parallel to the naphthalene plane was reduced significantly, corresponding to reduced π -electron delocalization. Please provide some explanation.
2. All samples formed spherical aggregates after spray drying, but SD-NBP(762202) and SD-NBP(750718) became nanosized flakes after electrodes fabrication. The morphology evolution during electrode fabrication needs to be investigated and discussed. In addition, the two-dimensional nanoflakes will necessarily lead to certain differences in XRD characterization. The ion diffusion pathway will also change, which may contribute to the fast-charging kinetics as well. These are not mentioned in the manuscript.
3. According to the presented data, SD-NBP(762202) delivered better rate performance and SD-NBP(750718) delivered better cycling performance. Please explain this phenomenon.
4. The cycling stability of SD-NBP(762202) and SD-NBP(750718) are not appealing in terms of 100 cycles. The morphology or crystal structure after cycling are expected to show the fading mechanism.
5. The reviewer is wondering why the bending vibrations perpendicular to the naphthalene plane and stretching vibrations parallel to the naphthalene plane are different in these samples. Does it associate with molecular ordering within the organic layer that may be also combined with crystalline analysis?
6. In a single π -stacking organic layer of SD-NBP(750718) or SD-NBP(762202), is there only one kind of molecules? The distinctive peak of (100) is identical to the molecule length, however, the crystallinity deteriorates after adding 3 components. Some comments are expected.

In addition, the following factors may be helpful to strengthen the presentation of this work.

7. Please provide more detailed interpretation about the machine learning section.
8. The summary part needs to be extended. Only two sentences are presented to conclude the major findings.
9. There are some typos in the maintext and figures. For instances, "we used a single phase naphthalene framework (SD-Naph) with Naph and Bph in a ratio of 75:25 (SD-NBP(750025))", "a temperature increase rate of 5 °C min. -1", etc. In Figure 1, "graphit" is a typo. The authors need to check the manuscript.

Reviewer #3 (Remarks to the Author):

- What are the noteworthy results?

-I find noteworthy the report on the fast charging capabilities of the iMOF material

and its role in enabling the reported two lithium charge transport at the interface (200 mAh/g reversible capacity).

-Also noteworthy in the report is the low process temperatures.

-The identification of the strain modes may help in the design of other MOFs for higher capacity.

● Will the work be of significance to the field and related fields? How does it compare to the established literature? If the work is not original, please provide relevant references.

-The work reports a stable relatively high voltage material in combination with choices from a machine learning technique.

● Does the work support the conclusions and claims, or is additional evidence needed?

-The work is thorough in experimental investigations. Raman studies are supported with theoretical first principles calculations so that possible modes are identified in the report. The rigidity of the carbonyl rings is reported in the study and upon strain utilized to aid in the fast charging.

-Cell cycling is reported up to 100 cycles and shows good retention in the 762202 samples.

-The machine learning methodology should be further clarified [in relation to ref 28], such as why the specific composition ratios were picked from the ternary diagram of Fig. S3. Does machine learning technique help to pinpoint the compositions optimized for high performance (voltage, capacity, hopping conduction) and low temperature treatment?

● Are there any flaws in the data analysis, interpretation and conclusions? Do these prohibit publication or require revision?

-In my opinion, the authors should further elaborate on several of these concerns prior to publication.

-I wonder about the polarization resistance of the materials and whether these resistances could be further reduced to help conduction mechanism.

-Could the material be prepared in different spray dry order so that the layering order is altered from that shown in Fig. 2? Are all samples fabricated in the order of ph, then Naph followed by Bph?

-I was not clear on the analysis presented in Fig. S9 and Table S.4 on C-rate per loading weight. Is the figure reporting the C-rate values?

-From the Raman analysis the peaks are shifted from theoretical values, indicating perhaps the length or size of the chains are somewhat different from expected. X-ray analysis suggests increased length along the plane a-c. What would be the packing limit of MOF crystal structure and is there possibility to have increased number of aromatic rings within a certain width (b-axis, for example)?

-Did the authors combine these materials as negative electrodes with positive electrodes such as NCM or LiFePO₄ or others? If so the authors may consider to

report on those results.

-Several typography errors are found throughout the text and in figure labels. For example Fig. S12 label "potential". The authors should make updates.

● Is the methodology sound? Does the work meet the expected standards in your field?

-Yes the results presented are thoroughly researched.

● Is there enough detail provided in the methods for the work to be reproduced?

-The methods section is detailed.

Response to the reviewers' comments (NCOMMS-22-29971-T)

Responses to the comments of Reviewer #1

We wish to express our appreciation to the reviewer for his or her insightful comments on our paper. The comments have helped us significantly improve the paper.

Comment 1: This manuscript reports an approach that combines machine learning and solution spray drying techniques to select and produce multivariate crystalline aromatic dicarboxylate intercalated metal-organic frameworks (iMOFs) for serving as Li-ion host negative electrode materials. To convince the concept, naphthalene-based multivariate iMOFs with specific compositions were predicted and prepared. These samples were characterized as nanoflake-like distorted crystals composed of π -stacked heteroaromatic multilayers and showed enhanced electrochemical properties, including high-rate capability and cycling stability, as compared to a single aromatic counterpart. Overall, this manuscript is in good shape, and the referee recommends acceptance of this paper in Nature Communications, but not before the referee's concerns listed below are addressed. The message delivered by Fig.1 was clear and straightforward, but Fig.1 also implies that the listed electrode materials are produced at a specific temperature which could be a bit misleading since some of these electrode materials are commercially produced at various temperatures. Taking graphite as an example, there is a huge difference in terms of the production temperature for natural graphite and artificial graphite. The referee would recommend the authors either improve Fig.1 with a temperature range instead of a single value of each material, or state clearly in the caption that these temperature values listed in Fig.1 are specific temperature conditions. The referee would also recommend moving Fig.1 to the Supplementary materials.

One last minor comment related to Fig.1 is that the term "Mxene (Ti₂AlC)" was used to describe MXene materials, however, this could be a bit misleading since Ti₂AlC is the MAX precursor to produce Ti₂CT_x MXene. And the temperature of Ti₂AlC was labeled as 1600 oC. This should be updated since many MAX phases, taking the well-known Ti₂AlC, Ti₃AlC₂, and Ti₃SiC₂ for example, can be produced at a temperature range of 1100-1400 oC (Ref: J Am Ceram Soc. 2021;104:659–690).

Response: We appreciate the reviewer's comment on this point. In accordance with the reviewer's comment, to avoid misinterpretation, we changed the treatment temperature of the electrode material, and then added that it is the representative specific temperature condition. Additionally, Fig. 1 was moved to Supplementary materials.

Comment 2: The authors should state the composition ratios of Naph, Bph, and Ph are depending on weight or molar.

Response: The composition of Naph, Bph, and Ph is based on molar ratios. We have added a description of molar ratios to the revised manuscript on p. 5 and 20, and in Supplementary materials.

Comment 3: More information should be provided on why these three particular ternary Ph-Naph-Bph compositions were selected by the machine learning prediction for further study. Based on what kind of parameters and conditions?

Response: We thank the reviewer for this comment. As shown in Supplementary Fig. 2, the crystal parameters of these three aromatic dicarboxylates have similar values for the lattice constant of the *b-c* plane, which corresponds to the direction of broadening of the tetrahedral LiO₄ network. This gave us the idea of a multivariate MOF sharing this *b-c* plane, as shown in the revised Fig. 1a. And, we incorporated predictions using machine learning, because there are many candidates with respect to the search for the optimal composition of the three-component system. Accordingly, We have added the following text as an explanation of the predictions made using machine learning (p. 30).

“The investigation of the optimal composition in the three-component organic linker was conducted by machine learning using a prediction model based on the random forest technique. The objective variable was reversible capacity, and the explanatory variables were similarity metrics as Pearson's correlation coefficient from the XRD pattern of arbitrary composition samples and that of each single phase such as Ph, Bph and Naph. The hyperparameters were optimized to show the lowest Root Mean Squared Error for the test data set, with accuracy improved by tenfold cross validation.”

Comment 4: The claim that “...a-axis oriented π -stacked heteroaromatic organic multilayers based on the naphthalene framework was the main component in SD-NBP(762202) and SD-NBP(750718)...” on Page 8 is one of the key features of these multivariate iMOFs samples, are there more experimental evidence, such as HR-TEM analysis, to support such claims? And on the same sentence, the authors continued, “...in addition to nanoflake formation due to the high specific surface area associated with the removal of water crystals.” What is the role of water in the multivariate crystallines? Some discussion would be appreciated.

Response: We strongly appreciate the reviewer's comment on this point. The proposed sample is difficult to observe the structure by HR-TEM as suggested by the reviewer because nano-sized flakes are formed by removing crystalline water by drying. In general, XRD patterns are often used as evidence for the confirmation for the multivariate MOFs (*Science* **2010**, 327, 846-850, *Inorg. Chem.* **2014**, 53, 5881-5883, *J. Am. Chem. Soc.* **2017**, 139, 14209-14216, and *J. Mater. Chem. A* **2015**, 3, 20145-20152). The XRD patterns for multivariate MOFs retain the same space groups even when multiple organic linkers are introduced. Therefore, we used the results of XRD patterns that can discuss the extinction law at $P2_1/c$ and crystal distortion as evidence for multivariate MOF in the proposed material. The analysis confirmed that the XRD pattern of each sample has a structure corresponding to the space group $P2_1/c$ of each organic linker. The 100 -plane peaks, which indicate *a*-axis regularity

as affected by the organic linker size, usually do not appear because the X-ray reflections are cancelled out by the extinction law in the space group $P2_1/c$ (*Cryst. Growth Des.* **9**, 2500-2503 (2009)). In contrast, we observed 100 -plane peaks corresponding to each organic linker in SD-NBP (762202) and SD-NBP (750718). This appearance of the 100 -plane peak suggests that the a -axis regularity changes while maintaining the crystal structure for the space group $P2_1/c$ of the respective organic linkers. Accordingly, we have added the following text (p. 7, 8):

“We performed an XRD comparison, one of the methods to confirm the multivariate MOF. According to Fig.1b, the XRD patterns after drying for each sample show that SD-Naph exhibits a bulk crystal pattern of the space group $P2_1/c$, whereas the heterogeneous samples exhibit a mixture of two phases, which are mainly component-rich. The respective 100 -plane peaks showing a -axis regularity depending on the organic linker size were clearly observed in SD-NBP(762202) and SD-NBP(750718), indicating the formation of organic layers by π -stacking interactions of each homoaromatic group rather than that of heteroaromatic mixtures.”

and

“suggesting a regularity change in the naphthalene framework that inhibits the extinction laws in the a -axis direction.”

In addition, the XRD patterns in SD-NBP (762202) and SD-NBP (750718) show crystal distortions that are not observed in the single organic framework. This suggests interference between crystals composed of their respective organic frameworks.

Furthermore, in accordance with the reviewer's comments, we have added XRD patterns before and after drying at 120°C to the Supplementary materials to discuss the effect of crystal water on the crystal structure. Changes in XRD patterns before and after drying were observed in SD-NBP (762202) and SD-NBP (750718), and the XRD patterns after drying displayed peaks originating from the space group $P2_1/c$. Therefore, we have added the following text (p. 6):

“The samples in SD-NBP (762202) and SD-NBP (750718) show changes in the X-ray diffraction (XRD) patterns before and after vacuum drying at 120°C that are not observed in the single phase SD-Naph (Supplementary Fig. 6), and the XRD patterns after drying displayed peaks originating from the space group $P2_1/c$.”

Comment 5: *What is the initial coulombic efficiency of these electrodes?*

Response: In accordance with the reviewer's comment, we have added the following initial charge and

discharge potential profiles in Li/sample cells with initial coulombic efficiency in Supplementary materials.

In addition, we have added the explanation of the above results (p. 9)

“initial coulombic efficiencies of 0.66 to 0.75 (Supplementary Fig. 11)”

Comment 6: On page 9, the authors wrote: “This result for SD-NBP (762202) implies that it can store more than 80% of its capacity at the 4C rate for 15 minutes of charging (Fig. 3f)...”. Did the authors test the SD-NBP (762202) electrode at the 4C rate to support such claims?

Response: We thank the reviewer for this comment. Due to the current density limitations of the electrode size and the lithium metal electrode utilization, measurements at the actual 4C rate cannot be performed. Therefore, we estimated the high rate based on the retention rate against C-rate, and as a result, SD-NBP(762202) was expected to retain more than 80% capacity at 4C-rate. Our original expression tended to be confusing. Accordingly, we have changed the following text (p. 10):

“The trend in the SD-NBP(762202) results implies the rate performance outlook is expected to be more than 80% of its capacity at the 4C rate for 15 minutes of charging.”

Comment 7: The authors used the term “during Li intercalation ” in captions of Fig.5 and Fig.S15 and Methods to describe the Raman results of these four electrodes. What are the lithiation states of the Raman measurements? If the Raman spectra were recorded based on fully lithiated electrodes, the

authors should state it as such.

Response: We wish to thank the reviewer for this comment. As suggested by the reviewer, the Raman measurements and calculation results are for each of the fully lithiated samples. Therefore, we have changed the following text (p. 17, 40):

“Raman spectra for each fully lithiated sample.”

Also, we have changed the following text in Supplementary Materials:

“Calculated and experimental Raman spectra for each fully lithiated state.”

Comment 8: *The authors claimed, “The Li intercalation in SD-Naph exhibits a flat potential profile with a slight potential hysteresis of approximately 15 mV (Fig. 6a)...” on page 14. Can the authors point out where this 15 mV has been observed?*

Response: We wish to thank the reviewer for this comment. Potential profile during Li intercalation in a Li/SD-Naph cell is shown to confirm the potential hysteresis.

Accordingly, we have added the figure in Supplementary Materials.

Comment 9: *On page 15, the authors wrote, “SD-NBP(762202) exhibits crystallographic distortions in the main Naph-based framework after Li intercalation, resulting in a reduced lattice size along the b axis, ...”. The author should make it clear that this “reduced lattice size” is under comparison with Li-intercalated SD-Naph rather than pristine SD-NBP (762202).*

Response: We thank the reviewer for this comment. In accordance with the reviewer’s comment, we have changed the explanation (p. 16):

“Li-intercalated SD-NBP(762202) exhibits crystallographic distortions in the main Naph-based

framework after Li intercalation, resulting in a reduced lattice size along the b axis, which corresponds to the p-stacking direction of naphthalene.”

Comment 10: The Summary on page 16 is too general; adding some specific results would make it more appealing.

Response: We wish to express our strong appreciation to the reviewer again for his or her valuable comments. In accordance with the reviewer’s comment, we have restructured our discussion by isolating extrinsic and intrinsic factors as mechanisms for the proposed material. The former and the latter are mainly attributed to the morphology factor by spray-dry synthesis and the electronic conduction factor by multivariate MOF, respectively. The morphological factor is confirmed by the CV scan rate dependence and Li diffusion coefficient results, while the electronic conduction factor is confirmed by the GITT potential behavior, the phase transition mechanism in the XRD patterns, and the molecular vibration suppression in the Raman results. Accordingly, we have modified our summary to reflect the factors and results as follows (p. 18-19):

“Our results reveal two fusion effects of extrinsic and intrinsic factors: control of nanoflake-like morphology formation in spray-dry synthesis and framework distortion in the optimal composition in multivariate MOFs using machine learning, respectively. The former effect leads to improved surface reaction and Li diffusion based on the results of the scan rate dependence in CV and GITT results, while the latter affects the avoidance of phase separation due to enhanced electron transfer, suggesting molecular vibration control effects as their mechanism based on the results of potential profiles, crystal structure changes during Li intercalation, and Raman spectra in lithiated samples. These effects resulted in faster charging performance, an expanded low-resistance capacity region, charge-discharge cycle stability, and improved high thermal stability.”

Thank you again for your comments on our paper. We trust that the revised manuscript is suitable for publication.

Responses to the comments of Reviewer #2

We wish to express our appreciation to the reviewer for his or her insightful comments on our paper. The comments have helped us significantly improve the paper.

Comment 1: In this manuscript, the authors fabricated the multivariate iMOFs electrodes to realize the reversible and fast Li charging, which are also unique anode materials with high safety. Optimal compositions were obtained based on machine learning. The spray-drying provided a synthetic solution with high efficiency and low energy cost. It is overall an interesting work, however, there are some obvious inconsistencies and major concerns that the authors need to take into consideration. Firstly, the relationship of structure-morphology-electrochemical properties is not strong. The NaPh, BPh and Ph alone have large resistance as electrode materials, but the multivariate iMOFs have small internal resistance and polarization. It seems that the intensity of stretching vibrations parallel to the naphthalene plane was reduced significantly, corresponding to reduced π -electron delocalization. Please provide some explanation.

Response: We wish to express our deep appreciation to the reviewer for his insightful comment on this point. As suggested by the reviewer, the specific molecular framework suppression in the multivariate MOF observed in the vibrational mode of the Raman spectra affects the electrochemical properties. The difference in the bending vibration corresponding to the perpendicular naphthalene plane means the structural fluctuation suppression leading to the persistent naphthalene planarity in SD-NBP(762202) and SD-NBP(750718). The bending vibration perpendicular to the naphthalene plane is a motion that disturbs the aromatic planarity, which negatively affects the electron transfer because the planarity is related to the delocalization of π electrons. The absence of the bending vibrations may be attributed to the strained structure observed in SD-NBP (762202) and SD-NBP (750718) and maintains π -electron delocalization by reducing these planarity-disturbing bending vibrations, thereby contributing to the enhancement of electronic conductivity during Li intercalation. Accordingly, we have changed the following text (p. 18):

“The former Raman spectral difference means the suppression of structural fluctuations leading to the persistent naphthalene planarity in SD-NBP(762202) and SD-NBP(750718). The bending vibration perpendicular to the naphthalene plane is a motion that disturbs the aromatic planarity, which negatively affects the electron transfer because the planarity is related to the delocalization of π electrons. The absence of the bending vibrations may be attributed to the strained structure observed in SD-NBP(762202) and SD-NBP(750718) and maintains π -electron delocalization by reducing these planarity-disturbing bending vibrations, thereby contributing to the enhancement of electron transfer avoiding phase separation during Li intercalation that leads to the observed fast charge performance with high temperature stability.”

Comment 2: All samples formed spherical aggregates after spray drying, but SD-NBP(762202) and SD-NBP(750718) became nanosized flakes after electrodes fabrication. The morphology evolution during electrode fabrication needs to be investigated and discussed. In addition, the two-dimensional nanoflakes will necessarily lead to certain differences in XRD characterization. The ion diffusion pathway will also change, which may contribute to the fast-charging kinetics as well. These are not mentioned in the manuscript.

Response: We wish to express our deep appreciation to the reviewer for his insightful comment on this point. We have considered the mechanism as two fusional effects of nanoflake-like morphology formation by spray-dry synthesis and multivariate MOF. These can be categorized as extrinsic factor related to Li diffusion path reduction and intrinsic factor related to phase transition mechanism due to strained crystal structure, respectively. The former effect is confirmed by the scan rate dependence in cyclic voltammetry and the Li diffusion coefficient results in GITT, indicating a surface reaction-dominated kinetics. The latter effect is confirmed by the results of potential profiles in GITT, Raman spectra and XRD patterns during Li intercalation, indicating the phase transition mechanism and its factors. Thus, we have changed our conclusions in the scan rate dependence in cyclic voltammetry to the following, which emphasizes that nanoflake-like morphology leads to surface reactions (p. 12-13).

“Furthermore, the anodic k_1/k_2 is strongly correlated with capacity retention at 2C for high rate performance in Fig. 2f (Fig. 4d), indicating that such kinetic change leads to the observed fast charging properties. This suggests that the kinetic behaviour in the above order of the series of materials shifts from the diffusion-limit reaction to the nondiffusion-limit reaction, indicating a fast surface reaction caused by the nanoflake-like morphology.”

In addition, we have added the following discussion of the phase transition mechanism in the R^2 value in GITT, used as a decision indicator for solid solution or two-phase coexistence reactions (p. 14):

“Thus, the R^2 values in SD-NBP (762202) and SD-NBP (750718) imply the solid solution reaction as phase transition mechanism.”

Comment 3: According to the presented data, SD-NBP(762202) delivered better rate performance and SD-NBP(750718) delivered better cycling performance. Please explain this phenomenon.

Comment 4: The cycling stability of SD-NBP(762202) and SD-NBP(750718) are not appealing in terms of 100 cycles. The morphology or crystal structure after cycling are expected to show the fading mechanism.

Response: We wish to thank the reviewer for this comment. In accordance with the reviewer's comment, we have added the following charge-discharge curves at low rates and XRD patterns before and after cycling. The peak intensity of the XRD pattern for the electrode state differs from that of the

powder in Fig. 1b due to the orientation of the sample. An unidentified broad peak, presumably due to side reaction, was observed around 22° at the SD-NBP(762202) electrode after the cycle test. The peak appearance may be related to the slight difference in cycling performance between SD-NBP (762202) and SD-NBP (750718). Although the XRD pattern after cycling showed a decrease in peak intensity compared to that before cycling, there were no significant changes in the charge-discharge curves at low rates and the peak positions in the XRD patterns before and after cycling, suggesting that such a decrease in peak intensity is due to the influence of the film formed by the reductive decomposition of the electrolyte during charge-discharge.

Comment 5: *The reviewer is wondering why the bending vibrations perpendicular to the naphthalene plane and stretching vibrations parallel to the naphthalene plane are different in these samples. Does it associate with molecular ordering within the organic layer that may be also combined with crystalline analysis?*

Response: We strongly appreciate the reviewer's comment on this point. We infer that this part pointed out by the reviewer corresponds to new findings in molecular vibrations and electronic properties inside the MOF, obtained by estimating vibrational models with phonon calculations and actual Raman spectral results. The discussion on the bending vibration is summarized in *Comment 1*.

Comment 6: *In a single π -stacking organic layer of SD-NBP(750718) or SD-NBP(762202), is there only one kind of molecules? The distinctive peak of (100) is identical to the molecule length, however, the crystallinity deteriorates after adding 3 components. Some comments are expected.*

Response: We thank the reviewer for this comment. As suggested by the reviewer, the multivariate

MOFs in the three components are poorly crystalline because of the broad peaks in the XRD patterns as shown in Fig. 1b. However, the peaks corresponding to the 100 -planes of each aromatic framework are clearly represented, and we consider the formation of organic layers composed of homoaromatic rather than heteroaromatic mixtures. Our original expression tended to be confusing. Accordingly, we have changed the following text (p. 7):

“The respective 100 -plane peaks showing a -axis regularity depending on the organic linker size were clearly observed in SD-NBP(762202) and SD-NBP(750718), indicating the formation of organic layers by π -stacking interactions of each homoaromatic group rather than that of heteroaromatic mixtures.”

Comment 7: Please provide more detailed interpretation about the machine learning section.

Response: We thank the reviewer for this comment. According to the reviewer’s comment, we have added the following text as an explanation of the predictions made using machine learning (p. 30).

“The investigation of the optimal composition in the three-component organic linker was conducted by machine learning using a prediction model based on the random forest technique. The objective variable was reversible capacity and polarization of charge and discharge related to internal resistance. The explanatory variables were similarity metrics as Pearson's correlation coefficient from the XRD pattern of arbitrary composition samples and that of each single phase such as Ph, Bph and Naph. The hyperparameters were optimized to show the lowest Root Mean Squared Error for the test data set, with accuracy improved by tenfold cross validation.”

Comment 8: The summary part needs to be extended. Only two sentences are presented to conclude the major findings.

We wish to express our strong appreciation to the reviewer again for his or her valuable comments. In accordance with the reviewer’s comment, we have restructured our discussion by isolating extrinsic and intrinsic factors as mechanisms for the proposed material. The former and the latter are mainly attributed to the morphology factor by spray-dry synthesis and the electronic conduction factor by multivariate MOF, respectively. The morphological factor is confirmed by the CV scan rate dependence and Li diffusion coefficient results, while the electronic conduction factor is confirmed by the GITT potential behavior, the phase transition mechanism in the XRD patterns, and the molecular vibration suppression in the Raman results. Accordingly, we have modified our summary to reflect the factors and results as follows (p. 18-19):

“Our results reveal two fusion effects of extrinsic and intrinsic factors: control of nanoflake-like morphology formation in spray-dry synthesis and framework distortion in the optimal composition in multivariate MOFs using machine learning, respectively. The former effect leads to improved surface reaction and Li diffusion based on the results of the scan rate dependence in CV and GITT results, while the latter affects the avoidance of phase separation due to enhanced electron transfer, suggesting molecular vibration control effects as their mechanism based on the results of potential profiles, crystal structure changes during Li intercalation, and Raman spectra in lithiated samples. These effects resulted in faster charging performance, an expanded low-resistance capacity region, charge-discharge cycle stability, and improved high thermal stability.”

Comment 9: There are some typos in the maintext and figures. For instances, “we used a single phase naphthalene framework (SD-Naph) with Naph and Bph in a ratio of 75:25 (SD-NBP(750025))”, “a temperature increase rate of 5 °C min. -1”, etc. In Figure 1, “graphit” is a typo. The authors need to check the manuscript.

Response: In accordance with the reviewer’s comment, the manuscript was carefully re-checked and revised.

Thank you again for your comments on our paper. We trust that the revised manuscript is suitable for publication.

Responses to the comments of Reviewer #3

We wish to express our strong appreciation to the reviewers for their insightful comments on our paper. We feel the comments have helped us significantly improve the paper.

Comment 1: · What are the noteworthy results?

-I find noteworthy the report on the fast charging capabilities of the iMOF material and its role in enabling the reported two lithium charge transport at the interface (200 mAh/g reversible capacity).

-Also noteworthy in the report is the low process temperatures.

-The identification of the strain modes may help in the design of other MOFs for higher capacity.

· Will the work be of significance to the field and related fields? How does it compare to the established literature? If the work is not original, please provide relevant references.

-The work reports a stable relatively high voltage material in combination with choices from a machine learning technique.

· Does the work support the conclusions and claims, or is additional evidence needed?

-The work is thorough in experimental investigations. Raman studies are supported with theoretical first principles calculations so that possible modes are identified in the report. The rigidity of the carbonyl rings is reported in the study and upon strain utilized to aid in the fast charging.

-Cell cycling is reported up to 100 cycles and shows good retention in the 762202 samples.

-The machine learning methodology should be further clarified [in relation to ref 28], such as why the specific composition ratios were picked from the ternary diagram of Fig. S3. Does machine learning technique help to pinpoint the compositions optimized for high performance (voltage, capacity, hopping conduction) and low temperature treatment?

Response: We wish to thank the reviewer for this comment. We have attempted to improve performance by multivariate MOFs in three components, which cannot be achieved by MOFs of a single organic linker. We chose the three organic linkers because, as noted in the manuscript in p. 5, they have a high degree of similarity in crystal structure other than the organic linker size, and thus we expected to achieve multivariate MOFs. However, since the optimal compositions among the three components were myriad and difficult to find, we searched for them using machine learning and found that high performance was achieved in specific compositions. In this study, electrochemical and spectroscopic analyses were performed to investigate the mechanism of the optimal composition obtained by machine learning.

Comment 2: Are there any flaws in the data analysis, interpretation and conclusions? Do these prohibit publication or require revision?

-In my opinion, the authors should further elaborate on several of these concerns prior to publication.

-I wonder about the polarization resistance of the materials and whether these resistances could be

further reduced to help conduction mechanism.

-Could the material be prepared in different spray dry order so that the layering order is altered from that shown in Fig. 2? Are all samples fabricated in the order of ph, then Naph followed by Bph?

Response: We appreciate the reviewer's comment on this point. A further reduction of resistance would be a multivariate MOF based on our concept, introducing an organic linker that forms a highly similar crystal structure, e.g., a heterocyclic aromatic linker. Our results are positioned to provide new directions for material design. The regularity of the stacking order of the π -stacked organic layers is not yet understood. The scheme shown in Fig. 2 is an image. Accordingly, we have changed the concept scheme in Fig. 1a as follows:

Comment 3: I was not clear on the analysis presented in Fig. S9 and Table S.4 on C-rate per loading weight. Is the figure reporting the C-rate values?

Response: We appreciate the reviewer's comment on this point. From papers reporting rate characteristics of MOF electrodes, the capacity at 80% retention relative to that obtained at low rates and its current density were investigated, and the C-rate at that time was obtained from the following equation (C rate = Current density (mA g^{-1}) / Reversible capacity (mA h g^{-1})). A large C rate value indicates a fast charging condition. Accordingly, we have added the above explanation in Supplementary Table 4.

Comment 4: From the Raman analysis the peaks are shifted from theoretical values, indicating perhaps the length or size of the chains are somewhat different from expected. X-ray analysis suggests increased length along the plane a-c. What would be the packing limit of MOF crystal structure and is there possibility to have increased number of aromatic rings within a certain width (b-axis, for example)?

Response: We strongly appreciate the reviewer's comment on this point. As pointed out by the reviewer, the difference between the experimental and calculated values can be attributed to the equilibrium volume and lattice volume, which are related to the bond lengths between atoms, and these

factors are influenced by the temperature and Li site occupancy in the calculation conditions. Accordingly, we have added the above explanation in Supplementary Fig. 25 (p. 26 in Supplementary Information).

Comment 5: -Did the authors combine these materials as negative electrodes with positive electrodes such as NCM or LiFePO₄ or others? If so the authors may consider to report on those results.

Response: We wish to thank the reviewer for this comment. In accordance with the reviewer's comment, to verify the effect of the proposed negative electrode materials as a device, the performance of an asymmetric capacitor combined with an activated carbon (AC) electrode was evaluated. Here, SD-NPB(762202) was compared to a single layer biphenyl (SD-Bph) electrode fabricated by spray drying, which exhibits the lowest resistance in the single-framework iMOFs. In this device, the negative electrode properties have a significant impact on the overall device performance because the electric double layer reactions with ion adsorption and desorption at the activated carbon positive electrode are sufficiently fast that negative electrode kinetics is the rate-limiting factor. Additionally, the overall thermal stability of the device also depends on the negative electrode performance. The negative electrode is pre-doped with Li to utilize the adsorption capacity of both anions and Li⁺ ions at the activated carbon positive electrode and the low-resistance capacity region at the negative electrode. In the device evaluation, we demonstrated an expanded low-resistance capacity area in the cell design, reduced internal resistance, and improved stability at elevated temperatures, as shown in the following figures.

Fig. 3. Device performances. a, b Design of an asymmetric capacitor with activated carbon (AC) positive electrode and range of capacity utilization due to internal resistance in SD-NPB(762202) (a)

and SD-Bph (b) negative electrodes with pre-lithiated treatment. **c-d** Initial charge-discharge curves (c), I - V resistance at 0.15 mA cm^{-2} (d), capacity retention after 1000 cycles at $20 \text{ }^\circ\text{C}$ and 1 mA cm^{-2} (e), and capacity retention at 0.15 mA cm^{-2} after storage at $60 \text{ }^\circ\text{C}$ (f) for each cell. The I - V resistance was calculated as the difference between the average voltage of charge and discharge (ΔV) divided by the applied current (I).

As mentioned in the introduction, SD-Bph exhibits large charge-discharge polarization at the end of Li de-intercalation, resulting in a design with a reduced capacity range, while SD-NPB(762202) can be designed with a wide capacity range (Fig. 3a, b). The asymmetric capacitors fabricated with each electrode showed charge storage reactions utilizing adsorption of both anions and Li^+ ions (Supplementary Fig. 15), and also confirmed both high capacity (Fig. 3c) and low resistance (Fig. 3d) in the SD-NPB(762202)-based cell.

Supplementary Fig. 15 Initial charge-discharge curves (a, c) and their differential capacitance dQ/dV plots (b, d) for SD-NPB(762202)/AC (a, b) and SD-Bph/AC (c, d) cells, respectively. The differential capacitance dQ/dV plots (Supplementary Fig. 15b, d) exhibited a typical butterfly-like shape suggesting electric double layer formation of both anions and Li^+ ions. This is evidence that the fabricated cell operates with the targeted reaction mechanism.

The SD-NPB(762202)-based cell displayed better capacity retention over 1000 cycles compared to that of SD-Bph (Fig. 3e; Supplementary Fig. 16).

Supplementary Fig. 16 The charge-discharge curve changes at 1000 cycles (a, c) and their discharge capacity changes and charge-discharge coulombic efficiency (b, d) for SD-NPB(762202)/AC (a, b) and SD-Bph/AC (c, d) cells, respectively.

Furthermore, in the long-term storage characteristics at 60°C, the SD-NPB(762202)-base cell also exhibited favorable capacity retention of more than 70% at long duration of 300 hours, compared to 50% for the SD-Bph base cell (Fig.f; Supplementary Fig. 17).

Supplementary Fig. 17 a, b The charge-discharge curve changes at 20°C for SD-NPB(762202)/AC

(a) and SD-Bph/AC (b) cells at various times after storage at 60°C.

These results mean that the proposed multivariate MOFs not only exhibit low resistance and wide utilization performance, but also improved thermodynamic stability. Accordingly, we have added the above explanation for device performance (p. 10-12).

Furthermore, we have added the following text in the online method as a procedure for our additional results (p. 20 and p. 24-25).

“The activated carbon (AC) electrodes were also prepared by a dispersion of active material (90 wt%), carbon black (4 wt%), styrene butadiene rubber (5 wt%) and carboxymethylcellulose (1 wt%) in water onto Al foil. The loading weights of both electrodes were 3-4 mg cm⁻².”

and

“The evaluated asymmetric capacitors were examined using the same laminate-type cells composed of AC positive and pre-lithiated SD-NBP(762202) and SD-Bph negative electrodes with separator filled with the same electrolyte. Prior to galvanostatic charge–discharge measurements of the asymmetric capacitors, as shown in Fig.a, considering their respective low-resistance capacitance ranges, SD-NBP(762202) and SD-Bph electrodes were pre-lithiated by discharging laminate-type Li/sample cells up to 65% and 75% of total Li intercalation capacity, respectively. Asymmetric capacitors were then prepared using the respective electrodes taken from the cells and the AC electrodes in an argon-filled glove box. The cells were cycled between 1.5 and 3.4 V at a current of 0.15 mA cm⁻² corresponding to the fully charged capacity of the cell per 2 h (0.5C in SD-NBP(762202)/AC cell and 0.75C in SD-Bph/AC cell) to confirm their initial performance at 20 °C. The cycling test using the same cell was cycled 1000 times between 1.5 and 3.4 V at a current of 1 mA cm⁻² (3.3C in SD-NBP(762202)/AC cell and 5C in SD-Bph/AC cell). For high temperature storage performance, the cells were charged to 3.4 V at a current of 0.15 mA cm⁻² and then stored at 60°C for each specified time. After returning to 20°C, the charge and discharge capacities were checked between 1.5 and 3.4 V at a current of 0.15 mA cm⁻².”

Comment 6: Several typography errors are found throughout the text and in figure labels. For example Fig. S12 label "potential". The authors should make updates.

Response: In accordance with the reviewer’s comment, the manuscript was carefully re–checked and revised.

We wish to thank the reviewer again for his or her valuable comments.

REVIEWERS' COMMENTS

Reviewer #1 (Remarks to the Author):

The authors have addressed my concerns, therefore, I recommend the acceptance of this manuscript in its current form.

Reviewer #2 (Remarks to the Author):

The authors have addressed the raised concerns carefully. The manuscript is well presented. The reviewer is glad to support its publication.

Reviewer #3 (Remarks to the Author):

The authors have addressed concerns posed in the initial review, and the paper is sufficiently ready for dissemination.